# Social and structural determinants associated with the prevalence of sexually transmitted infections among female commercial sex workers in Dhaka City, Bangladesh

**Mahbuba Kawser**[1]*, **Md. Nazrul Islam Khan**[1†], **Kazi Jahangir Hossain**[2], **Sheikh Nazrul Islam**[1]

**1** Institute of Nutrition and Food Science, University of Dhaka, Dhaka, Bangladesh, **2** National Institute of Preventive and Social Medicine (NIPSOM), Mohakhali, Dhaka, Bangladesh

† Deceased.
* mahbubakawser@gmail.com

**Data Availability Statement:** All data can be found in the manuscript and Supporting information files.

## Abstract

Female commercial sex workers (FCSWs) bear higher rates of sexually transmitted infections (STIs) among key populations. The association of structural determinants and STIs among FCSWs was not at the forefront of research earlier in Bangladesh. This study examined how structural factors correlate with the prevalence of STIs at physical/social/economic/policy levels among FCSWs in Dhaka city. 495 FCSWs were screened for HIV, hepatitis B, and syphilis. Structural variables (Individual risks, high-risk sexual behaviors, work environments) were extracted from the previous multi-level study on FCSWs and analyzed in 2020 to determine whether macro/micro-structural factors were associated with STIs. The prevalence of STIs was 43.6% (95% CI: 39.1%-48). Most (n = 207/495) FCSWs were infected with Syphilis or Hepatitis B, only 1.8% had co-infection, and none was positive for HIV. Multiple logistic regression revealed that 'Individual risk' factors like age ($\leq$18 years, adjusted odds ratio = AOR = .28; 18.1–29.9 years, AOR = .57), years in the sex industry (<1 year AOR = .15; 1–5 years, AOR = .39), and condoms as contraceptives (AOR = 2.7) were significantly associated with STIs. Considering 'High-risk behaviors' like monthly coitus with regular clients (AOR = .33), performing no anal sex ever (AOR = .03), and consistent condom use (AOR = .13) were less likely to be associated with STIs (P<0.05), while the association of ever group sex with STIs reported to double (AOR = 2.1). 'Work environment' like sex on roads/parks/shrines/markets (AOR = 2.6) and ever HIV-testing (AOR = 2.5) were significantly linked with STIs. However, micro-level factors like experiencing forced sex in the past year (AOR = 1.79) and condoms collected from hotel boys (AOR = .34) were significantly associated with STIs in the 'Hierarchical- model' with increasing model-power. 'Micro-structural' determinants predominated over 'Macro/policy-level factors' and profoundly influenced STIs. FCSWs need comprehensive and integrated interventions to promote accurate condom use perception, eliminate risky sexual behaviors, and provide quality reproductive health care. Necessary steps at the policy level are urgently needed to decriminalize commercial sex work.

**Funding:** The authors received no specific funding for this work.

**Competing interests:** The authors have declared that no competing interests exist.

## 1. Introduction

Bangladesh's female commercial sex workers (FCSWs) remain at the center of large-scale commercial sex industries (Quasi-registered brothels, on and off-street sex settings like hotels and rented houses). This high-risk group maintains transactional sex with other key populations like People who inject drugs (PWID), Bi-sexual, and Hermaphrodites/Hijra. They also have overlapping relationships with other bridging populations, such as non-injecting drug users, migrant/transport workers, and commercial/non-commercial partners [1–6]. Recently, new AIDS cases have increased in Bangladesh, posing the country at greater risk as 53% (7400 out of 14,000) of HIV-positive detected cases remain missing. Dhaka appears vulnerable to an HIV epidemic, with the highest number of AIDS patients (2,572), mainly concentrated among male-PWID [1–3, 5, 7]. However, Bangladesh maintains a <1% prevalence of HIV infection among key populations and <0.1% in the general population [2, 5, 8–10]. Despite having non-transactional sex partners, one-third of PWID in Dhaka bought sex from FCSWs or men who have sex with men, and <40% used condoms [2]. Bangladesh may be on the brink of an HIV epidemic due to continuing high levels of sexually transmitted infection/STI alone among FCSWs, as most (77.0%) of them had at least one STI and exhibited potential high-risk sexual behaviors [5, 10–14].

A complex synergistic relation exists between STIs and HIV [15–20]; the odds of HIV infection are significantly elevated when associated with active syphilis [19, 20] or multiple STIs [16, 17]. Ulcerative STIs (Syphilis/cancroid/Herpes virus-2) increase the risk of acquiring/transmitting HIV infection in some populations more than non-ulcerative gonorrhea/chlamydia by 2–3 times [21–24]. Additionally, FCSWs have 13-times more relative risk for the acquisition of HIV compared to the general population [11, 25] due to an increased likelihood of being biologically susceptible, having higher rates of STIs/symptoms [5, 6, 12, 26], and lower STI/HIV knowledge [2–4, 14, 27–29]. Different high-risk sexual behaviors like no/inconsistent condom use, multiple-partnership concurrencies, group sex, and heterosexual anal intercourse [3, 12, 14, 27, 30–33], and poor sexual reproductive health-seeking behaviors are ubiquitously present among them [6, 14, 27–29]. Moreover, sex workers in Bangladesh experience socio-cultural, environmental, and structural vulnerabilities like dire poverty, gender inequality, illiteracy, higher internal-external migration, and cross-borders with high HIV-risk countries [2–4, 29, 34]. More importantly, the criminalization of sex work (Macro-policy vulnerability) can exacerbate the situation by accelerating the spread of STIs/HIV infection into the general population [1, 4, 5, 11, 29]. Following the changing global pattern of sex in the sex industry since the 1980s [3, 35], female sex workers in Bangladesh have entered the sex trade through coercion/violence/deceit, poverty/homelessness, and sex trafficking or self-motivation [4, 12, 14, 34]. After migrating to Dhaka from other regions of Bangladesh, most FCSWs were frequently raped/sexually exploited while working as child laborers/domestic workers in homes/factories. Ultimately, 'Survival-sex' is their only option, and they are becoming more stigmatized and marginalized [4, 12, 34]. Higher rates of bacterial STIs and symptoms remain prevalent among FCSWs in Bangladesh despite receiving HIV/STI prevention services [2, 5, 6, 26, 36].

Studies reported that disease acquisition among commercial sex workers was influenced by the interaction of multilayered structural factors [3, 11, 14, 37]. In recent years, for HIV/STI risk-reduction, prevention scientists have been targeting structural issues—exogenous to the individual's behavioral and biological factors [37–40]. Shannon and colleagues (2014) offered a 'structural HIV-determinants framework,' a complex-dynamic interplay of macro-level policies (Migration, stigma, criminalization-law), community organization, and work environment by promoting/reducing interpersonal factors of both CSWs and their commercial/non-commercial partners (Dyad-level risks/protections) [40].

In Bangladesh, population-based studies on STIs among sex workers are limited due to the lack of diagnostic facilities, budgetary constraints, and the illegal nature of sex work [14, 26]. Moreover, research on the hepatitis-B virus among FCSWs was conducted decades earlier [13, 41]. Few epidemiological studies explicitly considered structural determinants as risk factors for the acquisition/transmission of STIs/HIV [40]. More importantly, the roles of contextual-structural factors on STI in a resource-poor setting are not been addressed before in Bangladesh. Understanding the association of different structural characteristics with 'STI prevalence' may assist in understanding the causes behind the higher rates of STIs among FCSWs and taking proper measures for its alleviation at policy/intervention levels, which might also impact HIV risk. Given the importance of exploring contextual risk behaviors associated with STIs among this difficult-to-reach population, this study was conceived to understand better different macro/micro-structural factors at physical, economic, social, and policy levels.

## 2. Materials and methods

### 2.1 Study population and sites

**2.1.1 Study population.**   Four hundred ninety-five (n = 495) FCSWs (who agreed to give 5 ml venous blood) participated in this study. Structural variables like socio-demographic/individual risks, sexual networking and high-risk sexual behaviors, and venue/work environments were extracted from the previous multi-level study on FCSWs: "Female sex workers lifestyle, Immuno-nutrition, and infectious disease prevalence in Dhaka city." In 2020, structural variables were restructured from the lifestyle data set and analyzed to determine whether macro/micro-structural factors are associated with STIs.

**2.1.2 Study site and sampling.**   The study mentioned above was cross-sectional, carried out in 2015 among randomly selected 635 FCSWs (140 refused to give blood) who worked in hotels/roads/parks/cinema halls/shrines/shops/elsewhere. A multistage sampling technique was employed to collect data from seven hotels and three spots according to the probability proportional to size. Female sex workers were reimbursed 500 BDT for each participant. This study's details and sampling technique are described elsewhere [42].

### 2.2 Ethical approval and informed consent statement

The Faculty of Biological Science's ethical review committees approved the 'study protocol' at the University of Dhaka. After briefing all FCSWs about the perspective of the study, oral consent was obtained from each participant. On the very day of the interview, written permission was obtained by the participant's signs/printed thumb on the 'consent form.' According to the Helsinki Declaration, confidentiality was ensured during and after data collection.

**2.2.1 Conceptual framework of the study.**   Based on prior research on various structural factors of HIV/STI risk environment [37–40, 43, 44], a 'conceptual framework' of the study (Fig 1) was hypothesized, which explains how macro/micro-structural factors can be associated with STI prevalence among FCSWs. In the context of Bangladesh, authors conceptualized that destitute young women enter the sex industry due to internal migration (Macro-social), poverty (Macro-economic), or deception/others. These factors could directly influence sex solicitation places (Micro-physical) and downwardly influence 'STI prevalence' via individual factors (Micro-physical/behavioral). On the other hand, sex solicitation places, sexual networking, and high-risk sexual behaviors (micro-physical/social) can directly influence STIs. The parallel existence of forced sex/violence/police arrest (Micro-policy) could directly influence HIV testing, access to condoms (Micro-policies), and sex solicitation places. These factors may also profoundly influence STIs directly/indirectly through other factors/networks.

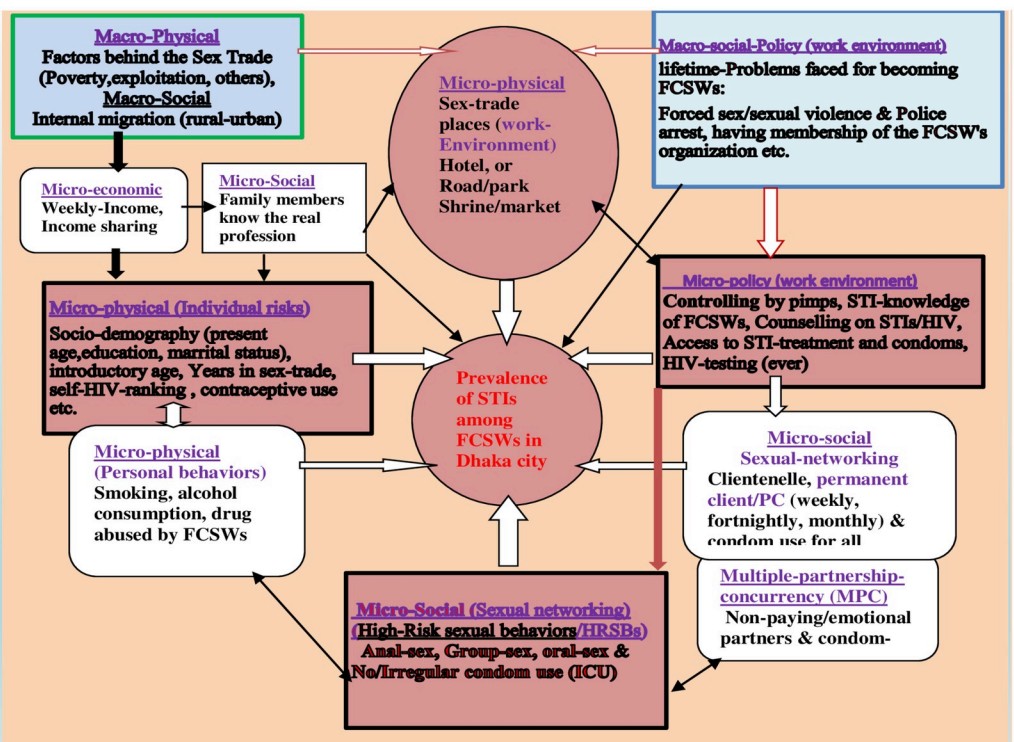

**Fig 1. 'Conceptual framework' shows different physical/social/economic/policy levels macro and micro-structural factors associated with the prevalence of sexually transmitted infections among female commercial sex workers (FCSWs) in Dhaka, Bangladesh.**

## 2.3 Measures

**2.3.1 Outcome variable.** The outcome variable was a positive test for Hepatitis B or Syphilis and the number of FCSWs living with co-infection (as no positive case of HIV was detected).

**2.3.2 Independent variables.** To determine the structural relationship with STI prevalence, the authors followed the 'HIV-risk environmental framework' proposed by Shannon and colleagues [39, 40]. Thus, extracted data were broadly segmented into three major domains under which macro/micro-structural factors remain.

*(1) Socio-economic and Individual risk factors.* Reasons for being sex workers, internal migration, seeking customers in another city, weekly income, and sharing income with others were socio-economic factors. Moreover, present age, education, marital status, and years in the sex trade, family members who knew about sex work, smoking, contraception, and self-reported HIV-risk were individual risk factors for FCSWs.

*(2) Sexual networking and high-risk sexual behaviors.* Current non-paying partner, Monthly coitus with regular clients, group sex ever, anal sex ever, last week's oral sex (fellatio/cunnilingus), and condom use were sexual networking and high-risk sexual behaviors.

**2.3.3 Estimation of 'consistent condom use'.** 'Consistent condom use' was estimated from positive answers to the 'seven questions' in the questionnaire. Participants were classified as engaging in consistent condom use if they reported that condoms were used during their

most recent encounters of vaginal, anal, and oral sex with all commercial and non-commercial sexual partners (Detailed in S1 Text).

*(3) Work environment.* Places of the sex trade, lifetime problems faced for being sex workers, last year's police arrestment and forced-sex, membership in FCSW's organization, controlled by pimps, condoms collection from, and HIV-testing ever (All categorical variables under the three domains mentioned above are detailed in S1 Text).

## 2.4 Laboratory methods

Viral screening of HIV and Hepatitis-B surface antigen (HBsAg) was performed by Enzyme-linked-immunosorbent-assay (ELISA) at 450 nm by using commercial enzyme-immunoassay/EIA kits (HIV: Bio-Rad Laboratories, USA; HBsAg: Omega-Diagnostics, UK) and EIA-readers (HIV: iMark, Bio-Rad, Japan; HBsAg: Labsystems, Multiskan-Ex). Positive and doubtful results were confirmed by both line-Immunoassay/LIA and immuno-chromatographic strip (Excel, China). Hepatitis B surface antigen (HBsAg) is a serologic marker representing acute/chronic hepatitis B infection. Venereal Disease Research Laboratory/VDRL test (Omega Diagnostics, UK) was used for the qualitative determination of syphilis, and Treponemal-Pallidum-haemagglutination/TPHA (Omega Diagnostic, UK) and rapid strips (Be sure, China) was used for the confirmation. Active syphilis was diagnosed if three consecutive test results were found positive. All laboratory analyses were completed at the 'Institute of Nutrition and Food Science, University of Dhaka.

## 2.5 Statistical analysis

Data were analyzed using a Statistical Package (SPSS Inc, Chicago, IL, USA, version 23.0), and normality tests (Kolmogorov–Smirnov/K–S goodness-of-fit) were performed before analysis. After screening and diagnostic confirmation of STIs, chi-square tests were performed in bivariate analysis to examine possible associations between STIs and structural variables. P-value $<0.05$ was set as significance. Variation inflation factors/VIFs were mostly one in the 'multi-collinearity test.' In multiple logistic regressions, the outcome variable was dichotomized as (1) for 'having STIs (syphilis/Hepatitis-B/both) and (0) for 'having no STI.' Relative odds ratios and 95% confidence interval were obtained to identify potential structural 'determinants' associated with STIs. Structural covariates found statistically significant ($P<0.05$) in bivariate analysis were introduced in the 'multivariable analysis' by 'Backward-Stepwise Elimination.' Additionally, 'Hierarchical-logistic regression' was performed after controlling the variables that remained in the adjusted model. Hierarchical logistic regression is a framework for model comparison of nested regression models. It helps evaluate the contributions of previously entered predictors or independent variables (IVs) as a means of statistical control and for examining incremental validity. If variables of interest explain a statistically significant amount of variance in the dependent variable (DV) after accounting for all other IVs. Moreover, Hosmer-Lemeshaw goodness-of-fit and Nagelkarke-pseudo-$R^2$ of the models were observed.

## 3. Results

### 3.1 Prevalence of STIs

The prevalence of STIs resulted in Table 1. ELISA indicated that none was positive for HIV. Prevalence of STIs among FCSWs (n = 216) was 43.6% [(95% CI: (39.1–48.0)] and 56.4% (n = 279) were free from infection. However, of them, 38.2% (n = 189) were afflicted with only

**Table 1. Prevalence of sexually transmitted infections (STIs) among female commercial sex workers (FCSWs) in Dhaka city, Bangladesh.**

| STIs | Female commercial sex workers | |
|---|---|---|
| | (n = 495) | |
| (n = 495) | Positive | No infection |
| | % (n) | % (n) |
| HIV/Human-Immunodeficiency-virus | Nil (000) | 100 (495) |
| Only Syphilis | 38.2 (189) | 61.8 (306) |
| Only hepatitis-B | 3.6 (18) | 96.4 (477) |
| Co-infection | 1.8 (09) | 98.1 (486) |
| Total (% n) (any STI/Co-infection) | 43.6 (216) | 56.4 (279) |
| Prevalence of STIs | 43.6% | |
| (95% Confidence Interval) (lower-upper) | (39.1–48.0)* | |

*Non-parametric test

syphilis, and 3.6% (n = 18) with only Hepatitis B. Overall, 41.8% (n = 207/495) had only one infection (syphilis/HBV), and 1.8% (n = 09) had co-infection.

## 3.2 Bivariate analyses

**3.2.1 Social and individual risk factors.** Bivariate analyses showed significant (P<0.05) associations between socio-economic/individual/behavioral factors with STI prevalence except for income sharing (Table 2).

Social factors: Internal migration to Dhaka city from other cities (macro-social) was very high (85.3%), and 55.8% entered the sex industry because of poverty (macro-economic), followed by exploitation (26.1%). Lower weekly income (<7000 BDT) was associated with higher prevalence (82.4%) and vice versa.

Individual risks: <30 years of age (referenced age ≥30), more schooling years, and unmarried status were associated with lower STI prevalence, while ≥6 years in the sex industry (referenced year ≥6) significantly increased the prevalence. Sex workers were more likely to be infected with STIs if family members knew about their professional job 'sex work". Interestingly, FCSWs who used condoms as contraceptives had a higher prevalence (49.5%) than pill users (21.8%). Low HIV risk perception was less likely to be associated with STIs (Table 2).

**3.2.2 Sexual networking.** Table 3 outlines different high-risk sexual behaviors associated with STIs. Having both non-paying partners and regular clients had a lower prevalence of STIs, and monthly coitus (≥ 3 times) with regular clients was reported to be the lowest (6.9%) prevalence than having none (P<0.05). All performed high-risk sexual behaviors (Group and anal sex, inconsistent/never condom use) were associated with STIs (P<0.05) except for the oral sex (fellatio/cunnilingus).

**3.2.3 Work environments.** Work environment variables influenced STI-prevalence (P<0.05) except for 'Memberships' of sex workers' organization and 'controlled by pimp' or Dalal (Bengali word, refers to pimps). 'Commercial sex' performed on roads/parks/shrines/markets/elsewhere was associated with a higher prevalence of STIs (81.5%) than in hotels (18.5%). Micro-social risk factors like lifetime-faced problems, last year's police arrest or forced-sex, condoms collected from hotel boys, and HIV testing (ever) were also associated with STIs (Table 3). Non-significant (P>0.05) factors (from Tables 2 and 3) in the bivariate analysis are presented in S1 Table.

**Table 2. Socio-economic and individual risk factors associated with STIs.**

| Structural- Environment (Macro/micro) | Structural determinants Socio-economic | Prevalence of STIs (%) n | | | P-trends |
|---|---|---|---|---|---|
| | | All (N = 495) | No (n = 279) | Yes (n = 216) | |
| Macro-physical | Reasons for being Sex workers | | | | |
| | Poverty | 55.8 (276) | 60.9 (170) | 49.1 (106) | P<0.001 |
| | Exploitation | 26.1 (129) | 23.7 (66) | 29.2 (63) | |
| | †Others | 18.1 (90) | 15.4 (43) | 21.8 (47) | |
| Macro-social | Migrated in from other cities | | | | |
| | Yes | 85.3 (422) | 82.4 (230) | 88.9 (192) | P<0.001 |
| | No | 14.7 (73) | 17.6 (49) | 11.1 (24) | |
| | Seeking customers in another city | | | | |
| | Yes | 14.7 (73) | 17.6 (49) | 11.1 (24) | P<0.05 |
| | No | 85.3 (422) | 82.4 (230) | 88.9 (192) | |
| Micro-economic | ‡Weekly income | | | | |
| | <7000 BDT | 72.1 (357) | 64.2 (179) | 82.4 (178) | P<0.001 |
| | ≥7000 | 22.8 (113) | 31.5 (88) | 11.6 (25) | |
| | $Iincome shared with others (forcefully/willingly) | | | | |
| | Yes | 28.1 (139) | 26.5 (74) | 30.1 (65) | P>0.05 |
| | No | 71.9 (356) | 73.5 (205) | 69.9 (151) | |
| | **Individual-risks/behavioral** | | | | |
| Micro-physical | ¶Present age (Years) | | | | |
| | ≤18 | 17.8 (88) | 26.9 (75) | 6.0 (13) | P<0.001 |
| | 18.1–29.9 | 46.3 (229) | 50.2 (140) | 41.2 (89) | |
| | ≥30 | 36.0 (178) | 22.9 (64) | 52.8 (114) | |
| | Education | | | | |
| | No schooling | 69.5 (344) | 59.5 (166) | 82.4 (178) | P<0.001 |
| | ¶¶1–12 schooling years | 30.5 (151) | 40.5 (113) | 17.6 (38) | |
| | Marital status | | | | |
| | Married | 41.6 (206) | 39.8 (111) | 44.0 (95) | P<0.001 |
| | Unmarried | 13.7 (68) | 20.1 (56) | 5.5 (12) | |
| | ¶¶¶Widowed/divorced/abandoned | 44.6 (221) | 40.1 (112) | 50.5 (109) | |
| | Years in the sex trade | | | | |
| | <1 | 10.7 (53) | 16.5 (46) | 3.2 (07) | P<0.001 |
| | 1–5 | 52.1 (258) | 62.4 (174) | 38.9 (84) | |
| | ≥6 | 37.2 (184) | 21.1 (59) | 57.9 (125) | |
| | Family members were informed about professional "sex work." | | | | |
| | Yes | 41.4 (205) | 34.1 (95) | 50.9 (110) | P<0.001 |
| | No | 58.6 (290) | 65.9 (184) | 49.1 (106) | |
| | Smoking | | | | |
| | Yes | 42.6 (211) | 38.7 (108) | 47.7 (103) | P<0.05 |
| | No | 57.4 (284) | 61.3 (171) | 52.3 (113) | |
| | Used Contraceptives | | | | |
| | Never | 16.8 (83) | 15.4 (43) | 18.5 (40) | P<0.05 |
| | Pills | 30.7 (152) | 37.6 (105) | 21.8 (47) | |
| | Condoms | 42.8 (212) | 37.6 (105) | 49.5 (107) | |
| | ¶¶¶¶Others | 9.7 (48) | 9.3 (26) | 10.2 (22) | |
| | Self-reported HIV-risks | | | | |
| | No | 52.3 (259) | 49.5 (138) | 56.1 (121) | P<0.001 |
| | Low | 27.9 (138) | 34.4 (96) | 19.4 (42) | |
| | High | 19.8 (98) | 16.1 (45) | 24.5 (53)() | |

†Anger on family members/husband/Drug-addiction/self-motivation/raped/tortured;

‡Did not work: (5.0%/n = 25);

$Local muscle men/family/police/others;

¶Mean-age:26.5±7.9;

¶¶Primary/1-5 Years (n = 129), Secondary/6-10 (n = 22), none for higher secondary (11–12 years);

¶¶¶Abandoned means when the husband left wife illegally;

¶¶¶¶Ligation/menopause/modern methods;

**Table 3. Sexual networking, high-risk sexual behaviors, and the work environment associated with STIs.**

| Structural- Environment (Macro/micro) | Structural determinants Sexual networking and Risky behaviors | STIs (%) n | | | P-trends |
|---|---|---|---|---|---|
| | | All | No | Yes | |
| Micro-Social | Having a current non-paying Sex Partner | 10.9 (54) | 13.6 (38) | 7.4 (16) | P<0.05 |
| | No | 89.1 (441) | 86.4 (241) | 92.6 (200) | |
| | Monthly coitus with regular clients | | | | |
| | 1–2 times | 20.2 (100) | 20.8 (58) | 19.4 (42) | P<0.05 |
| | ≥3 | 13.9 (69) | 19.4 (54) | 6.9 (15) | |
| | No regular clients | 65.9 (326) | 59.9 (167) | 73.6 (159) | |
| | Group Sex (ever) | 25.9 (128) | 20.8 (58) | 32.4 (70) | P<0.001 |
| | No | 74.1 (367) | 79.2 (221) | 67.6 (146) | |
| | Anal sex (ever) | 6.7 (33) | 3.6 (10) | 10.6 (23) | |
| | No | 93.3 (462) | 96.4 (269) | 89.4 (193) | P<0.001 |
| | Last week's oral sex/Fellatio | 12.3 (61) | 13.3 (37) | 11.1 (24) | P>0.05 |
| | No | 87.7 (434) | 86.7 (242) | 88.9 (192) | |
| | Last week's oral sex/Cunnilingus | 17.0 (84) | 19.0 (53) | 14.4 (31) | P>0.05 |
| | No | 83.0 (411) | 86.7 (226) | 85.6 (185) | |
| | Condom use | | | | |
| | †Never/inconsistent | 54.1 (268) | 41.9 (117) | 69.9 (151) | P<0.001 |
| | ‡Consistent condom use | 45.9 (227) | 58.1 (162) | 30.1 (65) | |
| | **Work environment** | | | | |
| Micro-physical | Places of the sex trade | | | | |
| | Roads/Parks/shrines/Markets | 64.0 (317) | 50.5 (141) | 81.5 (176) | P<0.001 |
| | Hotels | 36.0 (178) | 49.5 (138) | 18.5 (40) | |
| Micro-social | §Problems faced for being a sex worker (lifetime) | 81.2 (402) | 74.9 (209) | 89.4 (193) | P<0.05 |
| | No | 18.8 (93) | 25.2 (70) | 10.6 (23) | |
| | Police arrest (last year) | 34.7 (172) | 26.9 (75) | 44.9 (97) | P<0.05 |
| | No | 65.3 (323) | 73.1 (204) | 55.1 (119) | |
| | ¶Forced-sex (last year) | 47.3 (234) | 39.4 (110) | 57.4 (124) | P<0.001 |
| | No | 52.7 (261) | 60.6 (169) | 42.6 (92) | |
| Micro-policy | Membership in FCSW's organizations | 39.6 (196) | 39.8 (111) | 39.4 (85) | P>0.05 |
| | No | 60.4 (299) | 60.2 (168) | 60.6 (131) | |
| | Controlled by pimps/Dalal# | 7.1 (35) | 6.5 (18) | 7.9 (17) | P>0.05 |
| | No | 92.9 (460) | 93.5 (261) | 92.1 (199) | |
| | Access to condoms/Condoms collected from (n = 439) | | | | |
| | Colleagues | 23.0 (101) | 21.1 (54) | 25.7 (47) | |
| | Hotel boys | 29.2 (128) | 41.4 (106) | 12.1 (22) | P<0.001 |
| | Self-buying | 22.1 (97) | 15.6 (40) | 31.1 (57) | |
| | NGOs | 25.7 (113) | 21.9 (56) | 31.1 (57) | |
| | HIV-testing ever | 30.9 (153) | 23.7 (66) | 40.3 (87) | P<0.001 |
| | No | 69.1 (342) | 76.3 (213) | 59.7 (129) | |

†Never used (n = 56), inconsistent/occasional (n = 212);

‡condom use with every permanent client, non-paying partners for heterosexual vaginal intercourse, heterosexual anal intercourse, group sex, and oral sex;

§Police harassment/Rape/violence/physical abuse/people's nasty remarks/snatching money.;

¶Local Goons/police/customers/others;

#Dalal (Bengali word) means pimp.

### 3.3 Multivariable analysis

Multiple logistic regression (adjusted model-1) revealed that 'Individual risk' factors like age ($\leq$18 years, adjusted odds ratio = AOR = .28; 18.1–29.9 years, AOR = .57, Reference category ($^R$) $\geq$30), years in the sex industry (<1 year AOR = .15; 1–5 years, AOR = .39, ($^R$) $\geq$6 years), and condoms as contraceptives (AOR = 2.7, ($^R$) other methods) were significantly associated with STIs (Table 4, S1 Fig).

**Table 4. Structural determinants associated with STIs in multiple logistic regression models.**

| Structural determinants | Unadjusted odds ratio | Adjusted odds ratio (AOR) | |
|---|---|---|---|
| | | 95% CI (lower-upper) | |
| | †Unadjusted Model-1 | ‡Adjusted Model-1 | §Hierarchical Adjusted Model-2 |
| **Individual risks** | | | |
| *Micro-physical* | | | |
| Present age ($\geq$30$^R$) | | | |
| $\leq$18 years | .08 (.05-.19)* | .28 (.11-.68)* | .24 (.09-.61)* |
| 18.1–29.9 | .36 (.24-.54)* | .57 (.33-.98)* | .55 (.32-.96)* |
| Years in the sex trade ($\geq$6$^R$) | | | |
| <1 | .07 (.03-.17)** | .15 (.05-.43)** | .18 (.06-.56)** |
| 1–5 | .23 (.152-.341)* | .39 (.23-.66)** | .41(.23-.70)** |
| Family members informed about sex work (No$^R$) | 2.0 (1.4–2.89)** | | 1.62 (1.08–2.4) |
| Condoms as contraceptives (Others$^R$) | 1.3 (.64–2.26)* | 2.7 (1.2–6.1)* | 3.1 (1.3–7.4)* |
| Self-reported low HIV ranking (High$^R$) | .07 (.22-.40)** | | .47 (.23-.95) |
| **Sexual networking and high-risk sexual behaviors** | | | |
| *Micro-social* | | | |
| Monthly coitus with the regular client (No$^R$) | | | |
| 1–2 times | .76 (.484–1.19) | .23 (.152-.341) | .53 (.27–1.0) |
| $\geq$3 times | .29 (.158-.538)** | .33 (.13-.80)* | .37 (.15-.86)* |
| Performed group sex ever (No$^R$) | 1.8 (1.21–2.7)* | 2.1 (1.2–3.5)* | 1.9 (0.9–2.9)* |
| No heterosexual anal intercourse ever (Yes$^R$) | .31 (.15-.67)** | .03 (.04-.19)** | .06 (.02-.17)** |
| Consistent condom use (Inconsistent/Never$^R$) | .31 (.214-.453)** | .13 (.07-.23)** | .10 (.06-.19)** |
| **Work environment/sex trade places** | | | |
| *Micro-physical* | | | |
| Trading sex on the Road/Park/shrine (Hotel$^R$) | 4.3 (2.84–6.52)** | 2.6 (1.3–5.1)* | 2.1 (1.2–4.9)* |
| *Micro- social* | | | |
| Last year, police arrest (No$^R$) | 2.2 (1.52–3.23)** | | 1.79 (1.17–2.75) |
| Last year, forced sex (No$^R$) | 2.1 (1.44–2.97) ** | | 1.57 (1.03–2.38)* |
| *Micro-policy* | | | |
| Access to condoms/Condoms collected from (Colleagues$^R$) | | | |
| Hotel boys | .23 (.130-.436)** | | .34 (.130-.89)* |
| Self-buying | 1.6 (.933–2.87) | | 1.3 (.599-2.75) |
| Non-government organizations/NGOs | 1.1 (.683–2.0) | | 1.0 (.674–1.61) |
| HIV-testing ever (No$^R$) | 2.2 (1.48–3.2)** | 2.5 (1.5–4.3)** | 2.7 (1.5–4.8)** |

†Simple logistic regression.

‡**Multiple Logistic Regression**: Hosmer-Lemeshow goodness-of-fit chi-square/$x^2$ = 6.092, degree-of-freedom (d.f) = 8, P = .637; -2 Log likelihood (-2LL) = 451.989; Nagelkerke R-square = 49%; Overall-predicted-percentage = 76.8%

§ **Hierarchical Multiple logistic Regression**: Hosmer-Lemeshow goodness-of-fit/$x^2$ = 7.58, (d.f) = 20, P = .475; -2LL = 428.72, Nagelkerke R-square = 53%; Overall-predicted-percentage = 79.2%;

$^R$Reference category (parentheses);

*P<0.05;

**P<0.001

'High-risk behaviors' like monthly coitus with regular clients (AOR = .33, P = 0.036, Reference category($^R$) No), no anal sex ever (AOR = .03, P<0.001, ($^R$) Yes), and consistent condom use (AOR = .13, P<0.001, ($^R$) Inconsistent/never) were associated with STIs. In contrast, the association of group sex ever (P = .030) with STIs was reported to double as compared to no group sex ever (AOR = 2.1) (Table 4, S1 Fig).

'Work environment' like sex on roads/parks/shrines/markets (AOR = 2.6, P = 0.030, Reference category($^R$) Hotel), and HIV-testing ever (AOR = 2.5, P<0.001, ($^R$) No) were also more likely to be linked with STIs (S1 Fig). However, micro-social factors like 'last year's forced-sex' (AOR = 1.79, ($^R$) No) and if the source of the condom was a hotel boy as compared to colleagues (AOR = .34) were 66% less likely to be associated with STIs (both P<0.05) in the 'hierarchical-model' with increasing model-power (Overall predicted percentage increased from 76.8% to 79.2%) (Table 4).

## 4. Discussion

This study explored how macro/micro-structural factors correlated with the prevalence of STIs among female commercial sex workers (FCSWs) in Dhaka city. Although no HIV was found, a higher bacterial/viral infection prevalence among FCSWs is consistent with previous studies [5, 6, 9, 10, 12, 13, 26]. Donor support, government-led interventions, effective collaboration with implementing agencies, and peer education claimed to maintain a lower prevalence of HIV in Bangladesh [1, 2]. Moreover, the prevalence of hepatitis B is reportedly lower than in an earlier study [41]. However, participants of this study were not under any 'HIV/STI intervention' during the period of data collection, which could be the reason for the higher prevalence of syphilis. As reported by the recent surveillance [2], interventions like 'HIV prevention programs' and 'Harm reduction services' supported by donor agencies have been hampered since 2014. However, recently, bacterial STI prevalence was reported to decrease gradually after supportive 'HIV/STIs-intervention' [2, 36], although it remains high [6, 26].

This study revealed that micro-structural determinants (Fig 2) predominated over macro-factors for STI prevalence. It may be due to commercial sex industries submerged in a 'contradiction of laws' regarding the legitimacy of 'sex work' and 'sex workers.' As Bangladesh lacks macro/policy-level structural reformation regarding the legitimacy of 'sex work', FCSWs are

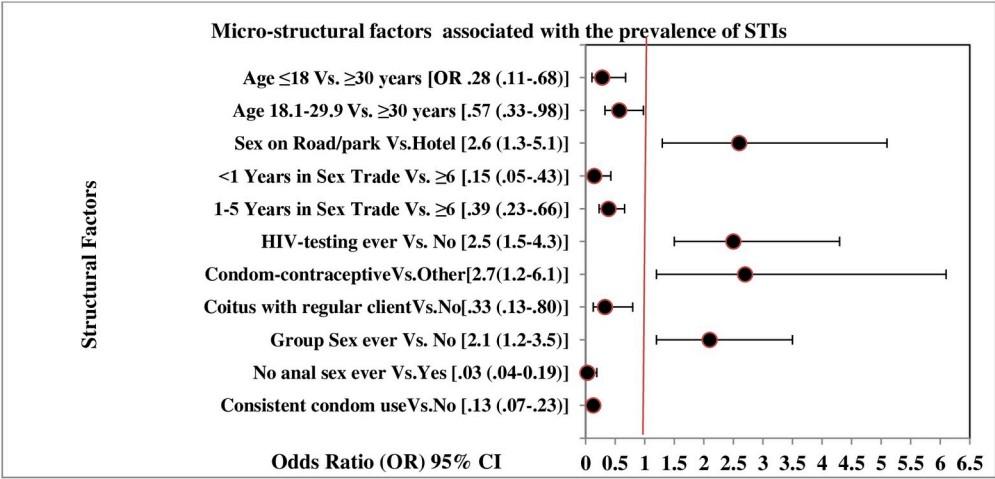

**Fig 2. Micro-structural factors associated with the prevalence of STIs among commercial female sex workers (FCSWs) in Dhaka City, Bangladesh.**

criminalized, often arrested under 'public-nuisance' or 'obscene' conduct provisions [1, 2, 4, 5, 29, 34]. Shannon and Colleagues (2014) reported that social, economic, and policy factors are more downstream products and interact with macro-structural factors (laws, stigma, criminalization) [40]. Structural HIV-determinants promote or reduce interpersonal factors (e.g., partner-level/dyad-level risks and protections, such as condom negotiation, sexual networks, and patterning) and interact with individual factors of sex workers, their clients and their intimate, non-paying partners, including behavioral (e.g., drug use, duration in sex work), biological (e.g., age, sex, race, HIV characteristics, STI co-infection), and host genotypic factors (e.g., host immunity) to shape HIV acquisition and transmission dynamics and epidemic trajectories at the individual and population levels [40]. A systematic review also outlined that only macro policy supporting-laws could lead to successful interventional outcomes [45] and HIV prevention and treatment care for CSWs in lower-middle income countries/LMICs [3, 11, 14, 37, 46]; otherwise, it diminishes promising health-promoting interventions [45].

## 4.1 Individual risk factors

In this study, various individual risk factors (Table 2) influence STIs profoundly; similar results are also found in other research [38, 44]. Young (<30 years) sex worker was 43%-72% less likely to be afflicted with STIs (P = .031) than aged one, which may be due to unprotected sex by older FCSWs or protected encounters by younger [17, 20, 47]. However, risky sex by under-aged/unmarried FCSWs is also reported [12, 48]. Years in the sex trade, a micro-physical factor positively associated with STIs (P<0.001) is also reflected in other studies [20, 49], while dissonance exists [16]. Surprisingly, using male condoms as contraceptives had a 2.7-times probability of getting STIs than pill-users (P = .046), which might be due to 'correct' (not before insertion/ejaculation) and 'consistent' (with commercial/non-commercial-partners) 'condom use' behaviors by pill-users, and incorrect and inconsistent condom behaviors by FCSWs who used condom as contraceptives. An earlier study [35] in Japan showed that taking oral contraceptives was associated with poor (inconsistent and incorrect) condom use and exposed vaginal sex workers (V-CSWs) to chlamydial infection. Reed et al. (2015) also reported that pills-users used condoms inconsistently [46], while another study by Kelly-Hanku et al. (2020) outlined that modern contraceptive method users had lower HIV infections than non-users [18]. Studies reported that most FCSWs treated 'condoms' as a means of 'contraception', not for the prevention of STIs/HIV [12, 47, 50], and few treated as dual-protection [51], have a potential unmet need for modern contraceptives (i.e., pill, intrauterine device) and family planning services [6, 12, 44, 46, 50]. Studies reported that female commercial sex workers [6, 12, 44, 46, 50], especially Asian FCSWs [6, 12, 44], did not meet their sexual and reproductive health (SRH) needs and experienced unwanted pregnancies, post-abortion, and pregnancy complications.

## 4.2 Sexual networking and high-risk sexual behaviors

The current study showed that high-risk sexual behaviors and steady relationships with regular clients (Table 3) strongly correlated with STIs among sex workers. However, comparing high-risk sexual behaviors of sex workers with other studies is difficult as they vary across countries, groups, and time-limit (lifetime-ever/monthly/past three months). Interestingly, monthly coitus with regular clients was a significant correlate of STIs (AOR = 0.33, P = 0.036) and showed protective roles on STIs, reflected in another study which showed not having a leading/casual-male partner had a twofold risk of HIV/syphilis [18]. However, higher monthly coital-frequencies with intimate partners were reported to be associated with less condom usage (P<0.05) [47] and increased risk of unprotected heterosexual anal intercourse (HAI) [31].

The group sex prevalence of this study (Table 3) reported half that of another study conducted among female sex workers in Melbourne [32], while this high-risk sexual act exhibited a twofold risk (AOR = 2.1, P = 0.030) for contracting STIs among FCSWs. An earlier study described that 40–45% of adult on-street beggars of Dhaka city engaged in multi-partners heterosexual sex with on-street FCSWs, with <10% reporting condom use [9]. The provision of condom use is significantly decreased in group sex due to lesser control over the situation.

Abstinence from heterosexual anal intercourse (Table 3) is 97% less likely (AOR = .03, P<0.001) to be associated with STIs. Previous studies also reported similar findings [15, 17, 31, 52]. Anal sex enhances HIV/STIs among FCSWs, riskier than heterosexual vaginal intercourse for lower condom usage [17, 27, 31, 52]; accelerated further with condom-less sex and improper lubricant use behaviors [17, 27, 52]. Female sex workers practice anal sex because of intimacy/poverty/coercion/peer pressure and lack of knowledge/perception of the exacerbated risk of anal sex for HIV/STI transmission [27].

This study elucidated (Table 3) that inconsistent/condom-less sex was 7.7 times (inverting AOR = .13, P<0.001) riskier for inflicting STIs and aligns with most studies [14, 16, 17, 49, 53]. Studies in Bangladesh also reported inconsistent/condomless sex as a high-risk factor for a new STI case among FCSWs [6, 26]. Despite consistent condom use being the only effective strategy in preventing STI/HIV [6, 14, 39, 47], the highest reported condom use by FCSWs was with new/one-time clients, and the lowest use with non-commercial/intimate sex partners [31, 40, 49]. Sex workers poorly understand the mechanism behind 'consistent condom use' and lack perceived knowledge [28, 48]. Interviewing and analyzing the data of this study regarding consistent condom use/CCU, the authors noticed that the 'perception of consistent condom use' was very poorly understood by the FCSWs of this study (69.4% of them had no schooling). If they used a condom every six clients out of seven, they thought it to be enough for protecting against STI/HIV and did not even count one 'unprotected event'. At the same time, one unprotected event with the intimate partner/regular client may result in the transmission of STIs from self to the partners or one partner to other partners or the bridging population, thereby creating a complex convoluted 'web of transmission'. Recent studies also reported that condomless sex/inconsistent condom use is a higher risk factor for a new STI case among Bangladeshi FCSWs [6, 26].

## 4.3 Work environment

The sex trade venue/typology of sex workers (Table 3) is a critical micro-physical factor of STIs. 'Commercial sex' performed on roads/parks/shrines/markets had a 2.6-times probability of being infected with STIs than in hotels (P = 0.030). It might be the lower hierarchal status of on-street FCSWs reported worldwide [2, 4, 5, 9, 16, 19, 20, 47, 48]. On-street or floating FCSWs are reported to have a lower price per sexual encounter [2, 4, 16, 48], have longer exposure times, use condoms inconsistently, exhibit riskier sexual behaviors [33, 48], and may face more violence/abuse for being illiterate and destitute [15, 16, 42, 47, 48].

Fewer FCSWs in this study have been tested for HIV, reported lower than in other studies [18, 54]. It might be due to being asymptomatic for any STI, newcomers, or many years in the sex trade, as nowadays, cost-free HIV testing is accessible for key populations in Bangladesh [2]. Multiple regression depicted that HIV testing is 2.5 times (P<0.001) more likely to be associated with STIs, as observed in a recent study [18]. A prior study in Bangladesh reported that FCSWs with a previous history of STI-testing were 6.6 times more likely to have HIV-testing [54]. Recent behavioral surveillance [2] reported a lower response to HIV-testing among hotel-based FCSWs than on-street FCSWs. Furthermore, other studies

reported that on-street/poor/older/illiterate FCSWs in Bangladesh with less knowledge of HIV/STI were less likely to be tested for HIV due to stigma/discrimination and time constraints [2, 4, 12, 28, 54]. A recent review [45] reported that financial constraints, time constraints, and stigma/discrimination are barriers to HIV testing among FCSWs. Tokar *et al.* (2018) described that different macro (HIV-testing policies, criminalization), meso (venues, social networking, stigma/discrimination, transport/costs), and micro-structural (socio-demography, knowledge/risk-behaviors) factors influence 'HIV-testing' [45]. Studies [37, 45] outlined macro policies (elimination of sexual violence) and micro social supports from family/peers/venue managers/healthcare workers can minimize sexual violence, facilitate CCU, avert HIV-infection [37, 39], and increase HIV-testing uptake among FCSWs [45].

## 5. Conclusion and recommendation

Micro-structural determinants were predominately associated with the prevalence of STIs among FCSWs. At the same time, the micro-social policy factor 'forced sex' and 'condom sources' were significantly associated with STIs only in the 'hierarchical model.' Comprehensive, integrated interventions for both groups of FCSWs (on-street or floating and Hotel-based) are warranted for accurate condom use perception. Especial lessons on the concept of 'condom use for dual protection' should be incorporated into the counseling session to understand condom use perception better. Moreover, counseling for minimizing risky sexual behaviors, enhanced consistent condom use behaviors, and ensuring quality 'Sexual and Reproductive Health' (SRH) care facilities are urgently recommended, mainly promoting modern contraceptive methods alongside 'HIV prevention programs' among FCSWs. As an STI prevention measure, a special 'social safety net' or rehabilitation program should be introduced for aged and experienced sex workers working longer in the sex industry. All FCSWs, especially hotel-based FCSWs, should be counseled or encouraged to test for HIV, as recent surveillance [2] reported lower response to HIV-testing among hotel-based FCSWs than on-street FCSWs. Necessary steps should also be taken to improve the economic stability of FCSWs (especially for on-street FCSWs) and decriminalize sex work at policy levels. More importantly, violence, a macro structural factor, is vividly common among all types of commercial sex workers across the globe [4, 5, 12, 29], underscoring the need for comprehensive policy reformation to avert all kinds of violence among FCSWs.

### 5.1 Strengths and limitations of the study

The main limitation of this study is its cross-sectional design, which limits causal inference. Despite FCSWs being screened for viral STIs like HIV and Hepatitis B and bacterial STI syphilis, other bacterial STIs (*Chlamydia* and *Gonorrhea*) were not tested, which is undoubtedly a limitation of this study. Moreover, information about condom slippage/breakage was not considered. On top of that, an association of structural determinants with STIs was not assessed earlier in Bangladesh despite having a higher STI burden among FCSWs. As data were randomly collected from the working places of FCSWs, they are more likely to be free from the phenomenon of 'regression to the mean' and 'selection bias' [43]. However, in the multi-level study on sex workers [42], one hundred forty (n = 140) FCSWs refused to give blood samples, which might be a selection bias. Moreover, reporting and recall biases for high-risk sexual behaviors cannot be ruled out as they are self-reported and highly personal. Nonetheless, consistent condom use was estimated whenever FCSWs asserted all seven answers of the corresponding 'condom use behaviors'.

## Supporting information

**S1 Text. Contains additional details on the different 'categories' of independent variables and estimation of consistent condom use (CCU) (Materials and Methods section).**
(DOCX)

**S1 Table. Bivariate analysis (Unadjusted odds ratio/UOR) shows different structural determinants associated with the prevalence of STIs among female sex workers.**
(DOCX)

**S1 Fig. Micro-structural factors associated with the prevalence of STIs among commercial sex workers in Dhaka city, Bangladesh.**
(XLSX)

## Acknowledgments

We mourn and recall the late Professor Md. Nazrul Islam Khan died of COVID-19 on April 18, 2021 (he led and supervised the main study). Authors are grateful to Parveen Begum, PhD, Consultant Nutritionist at 'Gulshan Ma O Shishu Shasta Kendra' and Shah Muhammad Anayetullah Siddiqui, Principal Scientific Officer at the Institute of Nutrition and Food Science (INFS), University of Dhaka, for their co-operation in analyzing the laboratory specimens. Authors are also acknowledging the contributions of Ms. Meherun Nessa, Associate Professor (English), Department of Language, Sher-e-Bangla Agricultural University, Dhaka-1207, and Jobaida Khanom, RDN, LDN, Renal Dietitian at Fresenius Medicare Care North America, USA for their patience, enthusiastic acts, and suggestions in copyediting this manuscript.

## Author Contributions

**Conceptualization:** Mahbuba Kawser, Md. Nazrul Islam Khan, Kazi Jahangir Hossain, Sheikh Nazrul Islam.

**Data curation:** Mahbuba Kawser, Md. Nazrul Islam Khan, Sheikh Nazrul Islam.

**Formal analysis:** Mahbuba Kawser.

**Investigation:** Md. Nazrul Islam Khan, Kazi Jahangir Hossain, Sheikh Nazrul Islam.

**Methodology:** Mahbuba Kawser, Md. Nazrul Islam Khan, Sheikh Nazrul Islam.

**Project administration:** Md. Nazrul Islam Khan, Sheikh Nazrul Islam.

**Resources:** Md. Nazrul Islam Khan, Kazi Jahangir Hossain, Sheikh Nazrul Islam.

**Supervision:** Md. Nazrul Islam Khan, Kazi Jahangir Hossain, Sheikh Nazrul Islam.

**Writing – original draft:** Mahbuba Kawser.

**Writing – review & editing:** Mahbuba Kawser, Md. Nazrul Islam Khan, Kazi Jahangir Hossain, Sheikh Nazrul Islam.

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
