## [Decision Letter · Decision Letter 0]

5 Apr 2022

PGPH-D-21-01081

Social and Structural Determinants Associated with the Prevalence of Sexually Transmitted Infections among Female Commercial Sex Workers in Dhaka City, Bangladesh

Dear Dr. Kawser,

Thank you for submitting your manuscript to PLOS Global Public Health. After careful consideration, we feel that it has merit but does not fully meet PLOS Global Public Health’s publication criteria as it currently stands. Therefore, we invite you to submit a revised version of the manuscript that addresses the points raised during the review process.

We look forward to receiving your revised manuscript.

Kind regards,

Seth Christopher Yaw Appiah,PhD PhD

Academic Editor

Journal Requirements:

1.Your co-authors:

Md. Nazrul Islam Khan -nik_infs@yahoo.com

Kazi Jahangir Hossain -jahangir1205@yahoo.com

Sheikh Nazrul Islam -sheikhnazrul09@gmail.com

,have not confirmed authorship of the manuscript. We have resent them the authorship confirmation email; however please check that the above email address for them is correct and follow up personally to ensure they confirm. 

Please note that we cannot proceed your manuscript  until we have received confirmations from all co-authors.

3. Please update the completed 'Competing Interests' statement, including any COIs declared by your co-authors. If you have no competing interests to declare, please state "The authors have declared that no competing interests exist". 

4. Please add a full list of legends for all supporting information files (including figures, table and data files) after the references list. 

5. We suggest you thoroughly copyedit your manuscript for language usage, spelling, and grammar. If you do not know anyone who can help you do this, you may wish to consider employing a professional scientific editing service.

Additional Editor Comments (if provided):

I find the article relevant both for the journal scope and its readership and in response to a growing Key population whose health needs need prioritization. As already indicated in the reviewer comments, authors would need to have a through grammar and syntax review of the paper to provide succinct reading and clarity of thoughts . A major review of all the required critiques from the reviewers is required after which it would be ready for acceptance

Regards

Dr Dr Seth Christopher Yaw Appiah

Reviewers' comments:

Reviewer's Responses to Questions

**Comments to the Author**

1. Does this manuscript meet PLOS Global Public Health’s publication criteria? Is the manuscript technically sound, and do the data support the conclusions? The manuscript must describe methodologically and ethically rigorous research with conclusions that are appropriately drawn based on the data presented.

Reviewer #1: Yes

Reviewer #2: Partly

2. Has the statistical analysis been performed appropriately and rigorously?

Reviewer #1: Yes

Reviewer #2: I don't know

3. Have the authors made all data underlying the findings in their manuscript fully available (please refer to the Data Availability Statement at the start of the manuscript PDF file)?

Reviewer #1: Yes

Reviewer #2: Yes

4. Is the manuscript presented in an intelligible fashion and written in standard English?

Reviewer #1: Yes

Reviewer #2: No

5. Review Comments to the Author

Reviewer #1: Review report.

Social and structural determinants associated with the prevalence of sexually transmitted Infections among female commercial sex Workers in Dhaka City, Bangladesh.

Summary

Female commercial sex workers are one of the key populations who are at risk of acquiring sexually transmitted diseases (STDS) and transmitting them to their sexual contacts who may also transmit STDS to other people in their sexual networks. This study looked at how macro/micro structural factors relate to the prevalence of STDS among female commercial sex workers in Dhaka at the physical, social, economic, and policy levels.

As expected, this research found a high prevalence of STDS among female commercial sex workers. However, despite the fact that female commercial sex workers had a higher prevalence of STDS (216/495), none of them tested positive for HIV which may be attributed to the low prevalence of HIV in Dhaka and Bangladesh in general. About 41.8 percent (n=207/495) had only one infection (syphilis/or HBV), and a small percentage (n=09/495) had both.

Female commercial sex workers require affordable, accessible, sexual workers’ friendly and comprehensive sexual reproductive health care services to reduce STDS prevalence in this population as well as spread to other people in their sexual network.

Comments and recommendations

The title and aim of the study are clearly stated. Their introduction and discussion was in relation to previous studies in the same field e.g., Paul A, 2020 ( line 564), Bagley C et al, 2017 (line 574 ). It identifies the prevalence of STIS, and a number of factors associated with the prevalence of STIS among female commercial sex workers in Dhaka . Findings of this study were compared to previous research findings in the discussion. While this research appears to be informative and sound, I recommend the following:

In the abstract, kindly guide the reader by putting subtopics e.g., introduction, methods, results, and conclusion. It is easy to follow when it has subtopics. Indicate the number of female commercial sex workers who participated in your study and confidence interval of overall prevalence of STIS.

Additionally, the language needs editing in some parts of this manuscript e.g., in abstract (line 14), “About 41.8% (n=207/495) had any one infection (syphilis/or HBV)” .

Introduction

Line 42 and 45: “During the period of 2020, 919 people were diagnosed with AIDS and Dhaka has the highest number of AIDS-patients (2,572) among high-risk-districts of the country.” Are these (2,572) only AIDS patients or both AIDS and HIV patients? Do they have AIDS defining illnesses to be referred to as AIDS patients? I am asking because you follow your sentence with ….” Although, Bangladesh maintains <1% prevalence of HIV infection….”

Line 55 to 57: “Ulcerative STIs (syphilis/cancroid/herpes simplex virus-2) increase 3-times whereas non-ulcerative STIs increases 2-times risk of HIV-transmission”. Kindly write its reference.

Line 64 up to 72 : Too long sentence and it becomes unclear along the way. Split it into two sentences.

Line 94, line 376, line 125 etc.: Kindly, abbreviations do not start sentences. “FCSWs………….”

Line 117 : “ ………..and analysed in 2020 to find out how macro/micro structural-factors associated with the prevalence of STIs at physical, social, economic and policy levels”. Analyzed in 2020? Did you do the data analysis for this study in 2020? Or it is the primary data which was analysed in 2020? I think , it is the original research, which was analyzed in 2020, though I am not sure from the way you wrote it. Kindly rewrite it clearly.

Methods

Line 146: Among the microphysical variables, for education, you have illiteracy and 1-12, what is 1-12? For people like me who do not know the education system of Bangladesh, 1-12 may not be clear to us. Is it lower (primary) level, middle ( secondary) level, higher level of education (tertiary, university) level or all levels of education?

Analysis

Overall, your statistical analysis and models are strong, and your results are compelling. I recommend considering which models/results to highlight in this paper to make it easy for the reader.

Did you test for multicollinearity before building the regression model ? It is not stated anywhere in your manuscript whether you did it or not, yet it is good to check the issue of multicollinearity every time before we build regression models. If yes, please mention it.

Line 185: I recommend you consider chronology of analysis e.g. We calculated the descriptive statistics to report on………… or We calculated the prevalence of STI among female commercial sexual workers……………. Then, we used bivariate analysis…………. From your results section (line 219), you started by reporting the prevalence of sexually transmitted Infections (STIs) among female commercial sex workers (FCSWs) in Dhaka city, Bangladesh. Then you followed it with bivariate analysis (line 257 ) however in your analysis section (line 182 ) bivariate analysis and prevalence of sexually transmitted Infections (STIs) do not appear anywhere. Kindly, try to be chronological and consistent.

Results

Line 259: Would start this paragraph with this statement, bivariate analysis showed association between socio-economic, individual, and behavioral-risk-factors and STI-prevalence except for income-sharing.

Discussion

Overall, I think the discussion is good. I propose you add interventions to reduce prevalence of STIS among female commercial sexual workers for each of the variables which were significantly associated with prevalence of STIS. What changes could be made to reduce the prevalence of STIS at the different levels in your discussion?

Reviewer #2: CORRECTIONS/EDITING/REVIEW

Although, higher prevalence of STIs (216/495) was noticed among FCSWs, but none of FCSW was positive for HIV.

Should be:

Although, a higher prevalence of STIs (216/495) was noticed among FCSWs, but none of FCSW was positive for HIV.

Line 1 ... higher rates

Line 5 …The association of

Line 6 … at the forefront of

Line 7 and 8 … individual risks and behavioural factors WITHOUT hyphen?

Line 11 … macro/or micro… No / required

Line 14 … ditto. No / required

Lines 15 to 30 … need to check the syntax, grammar and too much use of hyphenation: macro-policy, micro-structural et cetera

Introduction and beyond: the excessive use of hyphenation continues

transactional-sex

key-populations/KPs

permanent-clients

Please review this.

Line 47 … men-sex-with-men (MSM) ?? should this be men-who-have –sex-with-men, with or without the hyphens??

Line 48 - 49 … Bangladesh may BE (??) on the 49 brink of an

I gave up after this and merely read the paper as presented.

Some sentences were very difficult to understand, for example:

Line 267 – 269

Interestingly, FCSWs used condom as 268 contraceptive had higher prevalence (49.5%) than who (32.0%) used pill/other modern269 methods (ligation/Nor-plant/copper-T/Injection)

Review conclusion:

I was very interested in this paper after reading the abstract.

However, it soon became clear that the paper needs major copy editing by the authors to improve the syntax, grammar, phrasing and, perhaps, to shorten the paper so that it is clear and succinct.

I had no problems with the concept of the paper, methods and data gathered.

I will leave decisive comments on the statistical analysis to those with better understanding of statistics than me.

I think the paper should be thoroughly revised and resubmitted.

6. PLOS authors have the option to publish the peer review history of their article (what does this mean?). If published, this will include your full peer review and any attached files.

**Do you want your identity to be public for this peer review?** For information about this choice, including consent withdrawal, please see our Privacy Policy.

Reviewer #1: **Yes: **MARY LUWEDDE

Reviewer #2: **Yes: **John Lwanda

---

## [Decision Letter · Decision Letter 1]

5 Jul 2022

PGPH-D-21-01081R1

Social and Structural Determinants Associated with the Prevalence of Sexually Transmitted Infections among Female Commercial Sex Workers in Dhaka City, Bangladesh

Dear Dr. Kawser,

Thank you for submitting your manuscript to PLOS Global Public Health. After careful consideration, we feel that it has merit but does not fully meet PLOS Global Public Health’s publication criteria as it currently stands. Therefore, we invite you to submit a revised version of the manuscript that addresses the points raised during the review process.

Reviewers have agreed that this study is of public health importance, and the authors have improved the manuscript based on comments in the first round of the review. However, I agree with the reviewers that the manuscript is not written in standard English. The authors might need help from a more experienced researcher or English native speaker to condense the paper and correct typographical and grammatical errors throughout the text.

We look forward to receiving your revised manuscript.

Kind regards,

Siyan Yi, MD, MHSc, PhD

Academic Editor

Journal Requirements:

1. Please update your online Competing Interests statement. If you have no competing interests to declare, please state: “The authors have declared that no competing interests exist.”

Additional Editor Comments (if provided):

Reviewers' comments:

Reviewer's Responses to Questions

**Comments to the Author**

1. If the authors have adequately addressed your comments raised in a previous round of review and you feel that this manuscript is now acceptable for publication, you may indicate that here to bypass the “Comments to the Author” section, enter your conflict of interest statement in the “Confidential to Editor” section, and submit your "Accept" recommendation.

Reviewer #1: All comments have been addressed

Reviewer #2: (No Response)

2. Does this manuscript meet PLOS Global Public Health’s publication criteria? Is the manuscript technically sound, and do the data support the conclusions? The manuscript must describe methodologically and ethically rigorous research with conclusions that are appropriately drawn based on the data presented.

Reviewer #1: Yes

Reviewer #2: Partly

3. Has the statistical analysis been performed appropriately and rigorously?

Reviewer #1: Yes

Reviewer #2: I don't know

4. Have the authors made all data underlying the findings in their manuscript fully available (please refer to the Data Availability Statement at the start of the manuscript PDF file)?

Reviewer #1: Yes

Reviewer #2: Yes

5. Is the manuscript presented in an intelligible fashion and written in standard English?

Reviewer #1: No

Reviewer #2: No

6. Review Comments to the Author

Reviewer #1: Much better than the first version.

For readability and consistency, the article should be edited and updated by an English native speaker. There is too much use of abbreviations in this manuscript and the reader gets lost along the way.

“ no HIV was found among FCSWs”, extract from the discussion. Can you explain or attach a reason why no HIV was found among FCSWs?

Reviewer #2: I think that this article has material of public health importance.

However, I found the article a bit too long, wordy and still badly written.

It needs stricter copy editing and shortening. Using simpler presented statistics and tables would do no harm either.

7. PLOS authors have the option to publish the peer review history of their article (what does this mean?). If published, this will include your full peer review and any attached files.

**Do you want your identity to be public for this peer review?** For information about this choice, including consent withdrawal, please see our Privacy Policy.

Reviewer #1: **Yes: **Mary Luwedde

Reviewer #2: No

---

## [Decision Letter · Decision Letter 2]

8 Jan 2023

PGPH-D-21-01081R2

Social and Structural Determinants Associated with the Prevalence of Sexually Transmitted Infections among Female Commercial Sex Workers in Dhaka City, Bangladesh

Dear Dr. Kawser,

Thank you for submitting your manuscript to PLOS Global Public Health. After careful consideration, we feel that it has merit but does not fully meet PLOS Global Public Health’s publication criteria as it currently stands. Therefore, we invite you to submit a revised version of the manuscript that addresses the points raised during the review process.

Unfortunately reviewer #2 was not available to assess your revisions, and we made a decision to invite a third reviewer. The reviewers feel that the manuscript still needs to undergo copyediting/proofreading. They feel that further details need to be provided within the methodology section, and that the discussion section needs further revision regarding the presentation and interpretation of results.

We look forward to receiving your revised manuscript.

Kind regards,

Alex Schaefer, PhD

Associate Editor

Journal Requirements:

Additional Editor Comments (if provided):

Reviewers' comments:

Reviewer's Responses to Questions

**Comments to the Author**

1. If the authors have adequately addressed your comments raised in a previous round of review and you feel that this manuscript is now acceptable for publication, you may indicate that here to bypass the “Comments to the Author” section, enter your conflict of interest statement in the “Confidential to Editor” section, and submit your "Accept" recommendation.

Reviewer #1: All comments have been addressed

Reviewer #3: (No Response)

2. Does this manuscript meet PLOS Global Public Health’s publication criteria? Is the manuscript technically sound, and do the data support the conclusions? The manuscript must describe methodologically and ethically rigorous research with conclusions that are appropriately drawn based on the data presented.

Reviewer #1: Yes

Reviewer #3: No

3. Has the statistical analysis been performed appropriately and rigorously?

Reviewer #1: Yes

Reviewer #3: No

4. Have the authors made all data underlying the findings in their manuscript fully available (please refer to the Data Availability Statement at the start of the manuscript PDF file)?

Reviewer #1: Yes

Reviewer #3: Yes

5. Is the manuscript presented in an intelligible fashion and written in standard English?

Reviewer #1: No

Reviewer #3: No

6. Review Comments to the Author

Reviewer #1: Thank you for addressing all the comments raised in the last review. The English is much better than the previous one. Though there are some statements which are not clear to the reader. You can do one more editing again.

Title- Kindly edit the title by removing capital letters in the middle of the sentence.

“High-risk behaviors' such as monthly coitus with permanent clients (AOR/.33), group sex ever (AOR/2.1), heterosexual anal intercourse ever (AOR/.03), and consistent condom use (AOR/.13) were also significant predictors of STIs”. Why do you say that “consistent condom use” is a high risky sexual behavior?

'Work environment' like sex on roads/parks/shrines/markets (AOR/2.6) and HIV-testing ever (AOR/2.5) also correlated with STIs (P<0.05). “HIV-testing ever” does not fit in the 'Work environment' group. A work environment is the setting in which you perform your job, therefore “HIV-testing ever” seems misplaced.

…….and micro-policy factor, ‘getting condoms from hotel boys’ (AOR/.34). ‘Getting condoms from hotel boys’ is not clear.

Socio-economic/Individual risk factors: Behind the sex work. What does “Behind the sex work” mean?

In data analysis, did you do multivariate analysis or multivariable analysis after bivariate analysis? From your description, you did multivariable logistic regression not multivariate. Kindly check and clarify this.

Reviewer #3: This is the 3rd version of a manuscript that aims to identify social and structural factors associated with prevalent STIs among female sex workers in Bangladesh. The authors have made a valiant attempt to respond to earlier critiques; however, additional revisions are needed. As I was not one of the former reviewers, I unfortunately have some new concerns regarding presentation and interpretation of results, as well as some concerns with language as indicated below.

Language. The authors took to heart comments from the reviewers and had their paper revised by a native English speaker. However, there are still numerous grammatical issues.

The authors have reduced the number of abbreviations but several still remain that are not commonly used (e.g., HRSB, GS, HAIs, CCU, SLRs, PCs). Please do not create your own abbreviations.

In several places, the language used is pejorative and stigmatizing (e.g., ‘male homeless street beggars’, page 18 and ‘lower tier FCSWs’ page 18 ). The authors are urged to use person-centered language that is no-stigmatizing.

Throughout the paper, language is used that infers causality (e.g., ‘predictors’), but this is a cross-sectional analysis so causal inferences cannot be drawn. The authors should refer to their associations as ‘correlates’. On page 14, change the term ‘correlated with’ to ‘associated with’.

In several places, the authors use the term ‘prevalence rate’. Prevalence is a proportion, not a rate. Please remove the word ‘rate’ from these sentences.

The authors use the phrase ‘permanent clients’ but permanence seems too strong. I suggest ‘regular clients’ instead.

The last sentence in the Abstract does not make sense as written.

Top of Page 6. Data were likely ‘reconstructed’ not ‘re-arranged’.

“Dalal” need to be more clearly defined.

Bottom of page 9. The sentence beginning with “When family members…” does not make sense.

Table 2. The heading ‘Behind sex work” does not make sense.

Methodology. The authors categorize several of their variables as macro or micro at various levels. However, in some cases these levels are inappropriate. For example, being forced to have sex in the last year is referred to as ‘macro-policy’ which does not make sense. Similarly, how is entering the sex trade through ‘poverty/deception/others’ classified as ‘macro-physical’? Poverty is a macro-economic factor whereas deception would be micro-social. In table 2, the phrase ‘micro-physical’ is beside “individual risks/behavioral’. Which is it? The authors are encouraged to use categories already assigned by Shannon et al (Lancet 2015), or Strathdee et al (Lancet 2010).

It is not clear how the authors could infer that qualitative data on syphilis could be used to indicate active infection, which relies on quantitative titers.

On page 9, under the heading “Individual Risk Factors” the authors include several variables that are social-structural and not intrinsic to the individual (e.g. internal migration).

Exact p-values should be reported in the Tables.

Table 4, the final multivariate model, includes several variables (e.g. group sex) that are not statistically significant. Why?

Discussion. The Discussion should not focus on variables that were not statistically significant in the final model.

This sentence does not make sense: “The successful implementation of ‘macro-level factors’ can create pathways to alleviate micro-level factors.”

Limitations should include mention of the fact that 140 FCSWs were excluded since they refused to provide lab samples, which is indeed a selection bias.

7. PLOS authors have the option to publish the peer review history of their article (what does this mean?). If published, this will include your full peer review and any attached files.

**Do you want your identity to be public for this peer review?** For information about this choice, including consent withdrawal, please see our Privacy Policy.

Reviewer #1: **Yes: **Mary Luwedde

Reviewer #3: No

---

## [Decision Letter · Decision Letter 3]

4 Jul 2023

PGPH-D-21-01081R3

Social and structural determinants associated with the prevalence of sexually transmitted infections among female commercial sex workers in Dhaka city, Bangladesh

Dear Dr. Kawser,

Thank you for submitting your manuscript to PLOS Global Public Health. After careful consideration, we feel that it has merit but does not fully meet PLOS Global Public Health’s publication criteria as it currently stands. Therefore, we invite you to submit a revised version of the manuscript that addresses the points raised during the review process.

Unfortunately none of the previous reviewers were available to provide comments and it was considered necessary to invite an additional reviewer to assess your revised manuscript. The reviewer has raised several remaining significant scientific concerns about the study that need to be addressed in a revision.

We look forward to receiving your revised manuscript.

Kind regards,

Miquel Vall-llosera Camps

Staff Editor

Journal Requirements:

1. May we please request copyediting of your submission.

Reviewers' comments:

Reviewer's Responses to Questions

**Comments to the Author**

1. If the authors have adequately addressed your comments raised in a previous round of review and you feel that this manuscript is now acceptable for publication, you may indicate that here to bypass the “Comments to the Author” section, enter your conflict of interest statement in the “Confidential to Editor” section, and submit your "Accept" recommendation.

Reviewer #4: (No Response)

2. Does this manuscript meet PLOS Global Public Health’s publication criteria? Is the manuscript technically sound, and do the data support the conclusions? The manuscript must describe methodologically and ethically rigorous research with conclusions that are appropriately drawn based on the data presented.

Reviewer #4: Partly

3. Has the statistical analysis been performed appropriately and rigorously?

Reviewer #4: I don't know

4. Have the authors made all data underlying the findings in their manuscript fully available (please refer to the Data Availability Statement at the start of the manuscript PDF file)?

Reviewer #4: Yes

5. Is the manuscript presented in an intelligible fashion and written in standard English?

Reviewer #4: No

6. Review Comments to the Author

Reviewer #4: Overall

This study purports to be the first to examine correlations between STIs and structural determinants in Bangladesh. The authors report some interesting findings. However, there are various elements of the methods, discussion and other parts of the manuscript that I believe require editing and/or clarification before the paper can be seriously considered for publication. I apologize to the authors, as I know they have already revised and resubmitted this manuscript based on feedback from a prior reviewer, and I know it is frustrating to have new concerns raised by a new reviewer. My detailed comments follow.

General

• Based on the literature I’m familiar with, it seems unusual for a paper focused on “sexually transmitted infections” to only include Hepatitis B and syphilis. I would suggest that the authors make it clear in the title and/or early on in the manuscript that the focus here is narrowly on these two infections. The authors should also provide a justification for why these 2 are the focus.

• I suggest that the authors use an equal sign rather than a backslash when reporting odds ratios (i.e., AOR=x rather than AOR/x).

Abstract

• Suggest moving the 3rd sentence about lack of prior research in Bangladesh up to be the second sentence.

• In the sentence that begins, “In total, 495 FCSWs…” the authors should say “screened for Hepatitis B and Syphilis” rather than STIs broadly.

• In the sentence that begins “Most FCSWs were inflicted…” the word “inflicted” implies they were deliberately infected, so I suggest another word.

• When reporting the multivariable logistic regression results, for age and years in sex industry, the authors should mention the reference group.

• In the sentence about high risk behaviors, the inclusion of “no anal sex” and “consistent condom use” is confusing, as these are the opposite of high risk. I suggest either reorienting these variables throughout the manuscript/analysis to be “anal sex” and “inconsistent condom use” or not calling these “high risk behaviors.”

• It is unclear what “monthly coitus with regular clients” means and compared to way. Is this at least monthly on average, within the past month, monthly vs. less frequent, monthly vs. more frequent? Please clarify.

• Instead of “last year’s forced-sex,” I suggest saying “experiencing forced sex in the past year” and removing the quotation marks.

• What does “condom source” mean and what is the reference group to which the AOR corresponds?

• In the last sentence, why are there quotation marks around “commercial sex work?”

Introduction

• I would suggest the authors consider restructuring the introduction to start with basic epidemiology of HIV/STIs in FCSW in Bangladesh, then introduce the structural determinants framework and then use that framework to present what is already known about drivers of STIs, etc.

• In the sentence that begins, “Recently, new AIDS cases…” when the authors say 53% of cases remain “missing,” do they mean “undetected?” It’s not clear what “missing” means.

• The sentence that starts, “Bangladesh may be on the brink…” is confusing and it is unclear how the second part of the sentence demonstrates that the country is on the brink of an HIV epidemic.

• In the last sentence of the first paragraph, please expand and explain how criminalization of sex work can accelerate the spread of STIs/HIV into the general population.

• Remove quotation marks around “criminalization of sex work.”

• In the last paragraph, the authors mention the “illegal nature of sex work.” It would be helpful to say whether sex work is fully criminalized, whether some aspects are legal, etc.

• There’s no need for quotation marks around “STI prevalence” and I would suggest changing this terminology as prevalence may have population-level implications when the manuscript is examining associations with STIs within individuals.

Materials and Methods

• I am confused about the study population. Were they participants of a previous study who were recontacted to participate in a new study? Or is there just one study with the present manuscript drawing on different data from that study than a previous manuscript did? Please clarify.

• While Supplementary File 1 offers some additional information about the measures used, there are some things that remain unclear even there. I also think many readers are unlikely to refer to that document so some additional details within the main manuscript are needed.

• Outcome variable: Based on my reading of the manuscript, the outcome variable would be better described as a positive test for Hepatitis B or Syphilis.

• Independent variables that are not clear based on current wording/level of detail: “behind the sex work,” “seeking customers to another city,” “income sharing.” Please briefly explain what these mean in the text.

• For sexual networking and high risk sexual behaviors some variables do not include a time period. Does “non-paying partner” mean currently, ever, within a particular period of time? What is the time period for coitus with regular clients and oral sex?

• I would move up the description of consistent condom use to the area where condom use measure is first mentioned.

• What does “lifetime problems” mean?

• Is forced sex by a particular perpetrator? In the Supplementary file, it says “forced sex/violence,” so please clarify what is included here.

• Estimation of consistent condom use: Suggest rewording this to something like “Participants were classified as engaging in consistent condom use if they reported that condoms were used during their most recent encounters of vaginal, anal, and oral sex with all commercial and non-commercial sexual partners.”

• Laboratory methods: I’m not sure that this level of detail is necessary, but if it was requested by a prior reviewer, I defer to them.

• What does “qualitative determination” mean?

• In the statistical analysis section, the sentence about VIFs, is this reporting results of a multicollinearity test or describing methods?

• Please explain hierarchical logistic regression as not all readers may be familiar with this.

• I suggest describing the conceptual framework much earlier, before the description of measures so that the measures can be clearly situated within the framework.

• All of the references used in the “conceptual framework” section seem to be from other settings. Is there any literature from Bangladesh that led to this hypothesized framework? If so, please add. If not, I suggest the authors state that the framework is based on work from other settings that is hypothesized to be applicable to Bangladesh as well.

Results

• In the paragraph just before Table 1, sentence that begins “Sex workers were more infected…” I do not think that the authors mean they were more infected and suggest changing to “more likely to be infected.”

• In Table 1, I do not understand what the p-values correspond to, as there does not appear to be any comparison or test, just a presentation of the prevalence of Hep B and syphilis.

• Table 2: p-values cannot equal 0, though they may be infinitesimally small. Please edit to be p<0.001.

• Table 2 and Table 3: For binary variables, I suggest rather than having 2 lines, one with the variable and one with “yes” or “no” (e.g., migrated in from other cities, did not seek customers in another city), only the first line is included in the table.

• For the education variable, is everyone necessarily illiterate if they never attended school and is everyone who ever attended school literate? It is confusing to me why the categories are illiterate vs. any schooling rather than no schooling vs. any schooling.

• In the sexual networking section second sentence, I think the word “partners” should be added after “non-paying.”

• Here again I have a question about the meaning of “monthly coitus,” but if it clarified earlier as suggested this may be fine.

• In the work environments section, why are quotation marks used around certain words and phrases?

• Table 3: Are the options under “access to condoms or condoms collected from” mutually exclusive? Couldn’t the respondent have gotten condoms from more than one source, or is the question specific to the last time they accessed condoms?

• Multivariable analysis: add “Table 4” in the first set of parentheses.

• Should include reference group for categorical variables like age and years in sex work.

• Various comments like those related to “high risk behaviors” and presentation of AOR apply to this section as well.

Discussion

• Overall, I think the authors could do a more thorough job of interpreting their results in the context of prior literature and the intervention implications.

• In the first paragraph, sentence beginning “However, participants of this study…” add the word “which” after “data collection.”

• Why have donor agency interventions been hampered?

• First line on page 17, it’s not clear what the authors are referring to when they write “contribute in both ways.”

• In the next sentence, please expand/explain on what is meant by and what evidence there is for ONLY macro policy supporting laws leading to successful interventional outcomes.

• In the discussion about contraceptive type, a number of questions arose for me. Could respondents choose more than one option? If not, how would a person answer who uses both respond? Also, I think the authors should dig a little deeper into the literature to help explain this finding of condom use as contraceptives being associated with STIs. The bivariate association is not significant, but once it is put into a model that also includes other variables, including consistent condom use, the significant association emerges. How can we interpret this and what are the implications?

• I do not understand the last sentence of this paragraph related to unmet reproductive health needs.

• Page 18, 3rd line: Do you mean less or more condomless anal sex? Please clarify.

• In the sentence that begins “An earlier study described that…” the word “indulged” has some potentially negative connotations. Suggest using “engaged” as a more neutral word.

• The meaning of the last sentence on page 18 is unclear. Are the authors taking this from their findings or prior literature and what is the purpose here?

• Work environment section page 19: The last sentence implies that FCSWs face violence because they are illiterate and destitute, which I don’t think is supported by the literature. Please rephrase and/or explain.

• I’d like to see more discussion about the finding around prior HIV testing being associated with STIs. What could explain this? Are FCSWs at high-risk of HIV/STIs more likely to get tested for HIV/other STIs but not Hep B and Syphilis so their Hep B/Syphilis might have gone undetected so this is really a reflection of risk rather than a true association between testing and infection? Or are they getting tested and diagnosed but having trouble accessing treatment? Or some other explanation? What are the implications?

• First line of page 20, what “both groups” are the authors referring to?

• I’d like to see some implications/recommendations more closely/directly tied to the study’s results.

• Additional limitations: only testing for Hep B and syphilis and not other STIs like chlamydia and gonorrhea certainly limits the ability to generalize to STIs more broadly.

Figure 2

• The numbers at the bottom of the chart are not readable and need to be adjusted.

• Please add a vertical line at the odds ratio of 1 so readers can easily see which ORs are above and below that.

7. PLOS authors have the option to publish the peer review history of their article (what does this mean?). If published, this will include your full peer review and any attached files.

**Do you want your identity to be public for this peer review?** For information about this choice, including consent withdrawal, please see our Privacy Policy.

Reviewer #4: No

---

## [Editor Report · Decision Letter 4]

18 Dec 2023

Social and structural determinants associated with the prevalence of sexually transmitted infections among female commercial sex workers in Dhaka City, Bangladesh

PGPH-D-21-01081R4

Dear Kawser,

We are pleased to inform you that your manuscript 'Social and structural determinants associated with the prevalence of sexually transmitted infections among female commercial sex workers in Dhaka City, Bangladesh' has been provisionally accepted for publication in PLOS Global Public Health.

Best regards,

Humayun Kabir

Academic Editor